# A sustainable yarn tension control technique for optimizing textile dyeing efficiency and water use

Rana Ahmed[1], Motaz Amer[iD][2*], Lina Ismail[iD][3]

1 Department of Electrical and Control, College of Engineering, Arab Academy for Science, Technology and Maritime Transports, Alexandria, Egypt, 2 Basic and Applied Science Ins., College of Engineering, Arab Academy for Science, Technology and Maritime Transports, Alexandria, Egypt, 3 Department of Industrial and Management Engineering, College of Engineering, Arab Academy for Science, Technology and Maritime Transports, Alexandria, Egypt

* motaz.amer@aast.edu

## Abstract

The textile dyeing sector contributes significantly to excessive water and energy use and environmental pollution. Managing yarn tension during the rewinding process is a crucial challenge in textile dyeing, as it has a direct impact on yarn quality, resource consumption, and overall efficiency. Traditional tension control techniques ignore how tension variation affects the production of wastewater and the need for chemical treatment. Using a digital signal processor (TMS320F28335), this study introduces a novel event-driven yarn tension control method for soft winding machines that maintains ideal, constant tension across fine, medium, and thick cotton yarn counts. The suggested method dramatically reduces tension variance, as demonstrated by experimental validation conducted at a leading Egyptian cotton yarn producer. This decreases yarn defects by 35% and allows for a 27% increase in machine speed without sacrificing yarn quality. Most significantly, liquor ratio data show that optimized tension management reduces water consumption during dyeing by 65–70%, thereby lowering chemical use and wastewater generation. Additionally, energy use dropped by almost 20%, improving process sustainability. This integrated strategy promotes the triple bottom line of social, economic, and environmental advantages and is consistent with cleaner manufacturing principles. The results provide a scalable methodology for environmentally friendly textile production, especially appropriate for developing nations with limited resources.

## 1 Introduction

Textile dyeing is the process of imparting color to yarns, fibers, or fabrics [1]. Its origins date back to 2900 BCE in Egypt, where ancient Egyptians believed that dyeing added a touch of beauty to their textiles [2]. At that time, only natural dyes extracted from animals, insects, plants, or minerals were used to produce a variety of colors [3]. Over the centuries, the practice of dyeing with natural dyes has transformed into

**Data availability statement:** All relevant data are within the paper.

**Funding:** The author(s) received no specific funding for this work.

**Competing interests:** The authors have declared that no competing interests exist.

synthetic dyes, driven by changes in culture, technology, and the environment [4]. The textile industry is one of the most globalized industries and plays an important economic factor in Egypt, contributing approximately 12% of the country's export earnings. Additionally, Egypt hosts Africa's largest and most productive textile clusters [5]. Although textile products are essential for human beings' everyday lives, the textile industry is rated as the fourth largest contributor to climate change and environmental degradation, according to the European Environmental Agency (EEA) [6].

## 1.1 Water and chemical consumption in textile dyeing

Textile dyeing is a resource-intensive process that heavily relies on water throughout various stages, including washing, bleaching, and dyeing [7]. Approximately 28 million tons of textiles are dyed annually, consuming over 50 billion cubic meters of freshwater [8]. Furthermore, water demand in the industry is expected to increase by 400% by 2050 [9]. In addition to its significant water consumption, the dyeing process relies heavily on chemicals, which impact both environmental footprint and human well-being [10–12].

In recent years, governments and environmental agencies have been driving more rigorous emissions regulations to promote sustainability awareness in the manufacturing sector. The United Nations framework convention on climate change in Egypt in November 2022 highlighted the urgent actions needed for the growing threats to climate created by the textile and leather sectors. Failure to meet the 2030 emissions target could result in rising sea levels, extreme heat waves, and severe rainfalls [13]. Growing environmental risks due to the inefficient use of resources and lack of sustainability awareness created the need for running sustainable production practices, commonly referred to as "Cleaner Production" in the industrial sector. Cleaner production aims to minimize the negative environmental impact, reduce risk to human health, and lower costs [14].

## 1.2 Yarn tension in the rewinding process: current challenges

The textile production process relies on textile fibers that are categorized into natural fibers (e.g., cotton) and man-made fibers (e.g., polyester) [15]. This study focuses on Egyptian cotton, specifically Giza 86 variety, known for its superior quality including luxurious feel, softness, and lustrous appearance [16]. Since cotton fibers have limited elongation and are prone to breaking under high stress, it is essential to optimize spinning processes by controlling tension and speed during yarn production in order to balance productivity and fiber integrity [17,18].

Cotton yarns are typically supplied in white wound on paper cones. Before the dyeing process, the paper cone must be replaced with a plastic tube using a soft winding machine. Maintaining consistent yarn tension with minimal variation during unwinding is critical for minimizing defects such as thick places, thin places, and neps, which significantly affect yarn quality and create production inefficiencies including excessive water usage [19]. Additionally, the dyeing process generates colored wastewater, which significantly contributes to environmental pollution and resource depletion. Manufacturers face the challenge of balancing yarn tension

control, dyeing effectiveness, and water conservation. Achieving this balance remains a key research gap. The dyeing process generates colored wastewater, contributing to pollution and resource depletion. Manufacturers face the challenge of balancing yarn tension control, dyeing effectiveness, and water conservation.

### 1.3 Research gap and study objectives

Although previous literature extensively explored various water treatment and chemical reduction approaches, limited research has examined water consumption reduction through optimizing yarn rewinding parameters. Previous findings found that yarn rewinding causes significant challenges related to tension control which affects both final product quality and color consistency. To date, no previous study has evaluated the combined effects of tension control on yarn quality, water consumption, and production efficiency for cotton yarns in an industrial setting. Consequently, this study addresses this research gap by proposing a controlled, data-driven yarn tension control approach for cotton yarn dyeing that quantifies impacts on water consumption, energy use, and product quality. The research adopts an explanatory case study methodology within Egypt's dyeing textile industry,

This paper is organized as follows: Sect 2 reviews the relevant literature on textile wastewater, cleaner production strategies, and tension control mechanisms. Sect 3 presents a real case study demonstrating the influence of soft winding tension on yarn quality. Sect 4 Proposes the event-driven yarn tension control technique with detailed system design. Sect 5 presents the methods and materials. Sect 6 presents experimental results, and discussion with statistical analysis. Finally, Sect 7 draws conclusions and recommendations for future work.

## 2 Literature review

### 2.1 Environmental impact and wastewater treatment in textile dyeing

The effluent discharge from the dyeing process is highly polluted, containing chemicals, dyes, metals, and other harmful substances that contribute significantly to environmental degradation and cause health risks to humans and animals [1,20–23]. As global environmental concerns grow, textile industries increasingly adopt cleaner production practices to align with sustainable development goals. Efforts have primarily focused on reducing resource consumption, particularly water consumption. Discharged wasted water from the dyeing process is colored and contains elevated levels of salts, Chemical Oxygen Demand (COD), Biological Oxygen Demand (BOD), Total Dissolved Solids (TDS), and various contaminants such as metals, nitrates, and chlorides [24].

An extensive review of textile wastewater treatment methods has categorized them into oxidation, physical, biological, and physicochemical approaches [25]. Another detailed study classified the water treatment methods into three main categories: (a) biological treatment methods, such as fungi, algae, bacteria, and microbial fuel cells; (b) chemical treatment methods, including photocatalytic oxidation, ozone, and Fenton's process; (c) physicochemical treatment methods such as adsorption, ion exchange, coagulation, and filtration [26]. Electrocoagulation is more effective than conventional biological treatment, advanced oxidation techniques, and chemical coagulation techniques [27]. Additionally, studies have reported sustainable benefits from the advanced oxidative process in terms of water, chemicals, and cost savings [28,29]. Chai et al. developed a fibrous membrane inspired by natural structure, demonstrating high efficiency in water treatment [30].

An essential aspect of wastewater treatment in textile dyeing is removing coloring matter before discharge into the aqueous ecosystem [31]. The decolorization process can be achieved through biological or non-biological methods. The selection depends on water pH, dye type, toxic compounds, and salinity [32–34]. A promising approach involves biotechnological techniques, which remove color from textile wastewater and enable the complete mineralization of dyes [35]. Other studies focused on reusing the treated dyeing wastewater [32,34,36,37]. Integrating a membrane bioreactor with reverse osmosis has been found to significantly reduce COD, BOD, TDS, and pH levels while effectively removing color. This facilitates the reuse of treated water in fabric washing and rinsing, contributing to more sustainable textile production [37].

Waterless dyeing technology is another area for attraction; for instance, Xu Suxin et al. [33] developed a new solvent dyeing method for polyester material, effectively eliminating water usage in the textile dyeing process. Innovative technology reduces water consumption and minimizes the need for chemical treatments and associated costs. While waterless dyeing technologies and the reuse of treated water can significantly mitigate environmental impacts, social awareness and the lack of technology know-how remain key barriers to limiting the implementation of cleaner production [38].

Consequently, implementing water management and tracking systems is essential for optimum water recovery and reuse. Baban et al. [39] integrated a water management system with the cleaner production approach, achieving a 75% reuse of wastewater, a reduction in pollutant concentrations for COD by approximately 80%, and 99% color removal.

## 2.2 Chemical usage and cleaner production practices in textile dyeing

As a significant amount of chemicals are used in the textile dyeing process, several approaches are found to optimize the amount of chemicals used and minimize or replace them. An integrated pollution prevention and control system was used to evaluate the chemicals used in a textile dyeing factory regarding their toxicological effects. The study revealed that 29 toxic chemicals should be replaced to obtain cleaner production [40,41]. The European commission has proposed an approach known as "Best Available Techniques (BAT)" to use natural resources efficiently and reduce the pollution loads in the production process of textile industries [42]. The implementation of BAT has shown remarkably successful results in terms of the consumption of water, chemicals, and energy, as well as emissions of gases. Furthermore, the payback period for such a program ranges from 3 months to 4 years maximum [41].

## 2.3 Yarn tension control and its impact on yarn quality and resource efficiency

Although previous literature on the textile dyeing process has extensively explored different water treatment and chemical reduction approaches, limited research has examined water consumption reduction by optimizing yarn rewinding parameters. Existing literature indicates that yarn rewinding poses challenges related to tension control, significantly affecting the final product's quality and color consistency. For instance, a study by Ozge Celik and Recep Eren [20] analyzed the effect of yarn length on yarn tension, suggesting that the findings could be applied to design a tension control system for obtaining optimum production efficiency. Another study [43] developed a fuzzy analytical hierarchy process to evaluate 16 barriers affecting the yarn winding process and proposed further research to identify the root causes of these barriers.

This final subsection covers yarn tension and its effect on defect types (thick, thin places, neps), quality impact, uneven package density effects on dye penetration and water consumption, prior studies on tension control. Thus, by concentrating on the optimization of yarn tension during the soft winding process—a crucial parameter that has been mainly disregarded despite its direct impact on yarn quality, water consumption, and overall production sustainability—this study seeks to close a significant research gap in the textile dyeing literature. Although water treatment and chemical reduction have been the subject of much research, little is known about how to control yarn tension upstream in order to accomplish these sustainability objectives. Yarn defects, colour uniformity, and resource efficiency are all impacted by yarn tension, a complex variable that is influenced by a number of variables, including yarn count, unwinding speed, and balloon length.

Tension variation resulted in non-uniform package winding, creating air pockets and variable density zones. Thus, during dyeing process, this non-uniformity necessitated extended dye bath contact times and in turn increased water volumes to achieve adequate dye penetration. On the other hand, uniform tension enables better dying penetration and reduces the liquor ratio. In conclusion, optimizing tension control during rewinding not only enhances yarn quality but also minimizes water consumption by ensuring uniform package density.

By customizing an event-driven tension control method for Egyptian cotton yarns in soft winding machines, this paper expands on the authors' basic work from a previous study that created a tension control mechanism especially for polyester yarns using a unique control approach [44]. In contrast to the earlier approach, the current study uses real-time digital

signal processing (through the TMS320F28335) to dynamically control tension. This reduces yarn imperfections, stabilizes rewinding speed, and achieves notable dyeing energy and water savings.

To the best of the authors' knowledge, no previous thorough method has addressed tension optimization and shown quantifiable economic and environmental benefits in the context of dyeing cotton yarn. Therefore, by directly connecting yarn tension control to water and energy savings, improving fabric quality, and promoting cleaner production principles—all of which are particularly pertinent for resource-constrained textile manufacturing settings worldwide—this study fills a critical gap.

## 3 Case study: Influence of soft winding tension on yarn quality

EL Sharq El Awsat Co., established in 1917, is one of Egypt's leading textile manufacturers specializing in the dyeing of cotton yarns for ready-made garment production. The company plays a significant role in Egypt's textile industry, known for producing high-quality dyed yarns that cater to both domestic and international markets. With a commitment to modernizing its manufacturing processes, EL Sharq El Awsat Co. embraces innovative techniques to improve product quality, enhance production efficiency, and adopt sustainable practices, making it an ideal setting for piloting advanced yarn tension control technologies as part of cleaner production initiatives.

The crucial preparatory steps in the textile dyeing process at EL Sharq El Awsat Co. are depicted in Figs 1 and 2. The two types of yarn packages used before and after rewinding are contrasted in (Fig 1), the rewound yarn on plastic tubes and the raw yarn supplied on paper cones. Cotton yarn is first delivered from spinning in the form of raw paper cones, but because of their fragility and incompatibility with dyeing equipment, these cones are not suitable for dyeing. Thus, the yarn is transferred from the paper cones onto stronger and more dye-compatible plastic tubes using a soft winding machine, as illustrated in (Fig 2). In order to guarantee uniform yarn package density, which has a direct impact on dye penetration, fabric quality, and resource consumption, this rewinding process must maintain constant and regulated yarn tension. Yarn packages that have been properly rewound on plastic tubes offer a consistent, stable foundation for effective dyeing, lowering flaws and water usage in later steps.

The quality of the yarn and the effectiveness of the subsequent dyeing process are both significantly impacted by variations in yarn tension during the rewinding process. Defects like thick spots, thin spots, and neps are caused by inconsistent tension, which reduces yarn evenness and speeds up breakage. Uneven package density brought on by these flaws

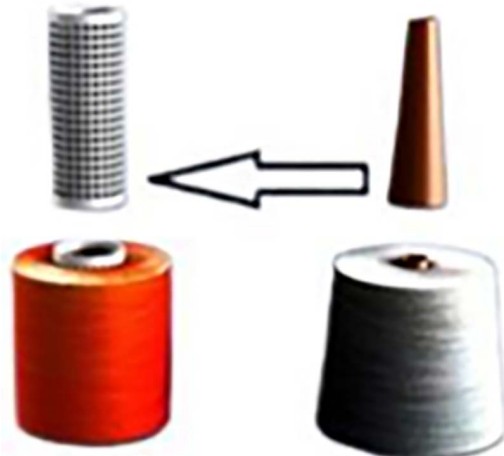

**Fig 1. Plastic tube versus raw paper cone [Courtesy of EL Sharq El Awsat Co.].**

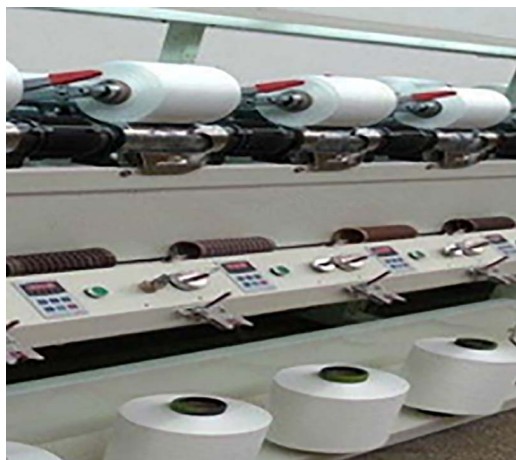

**Fig 2. Soft winding machine.**

results in uneven colour shades and poor dye penetration. Manufacturers frequently use a lot more water to make up for this, which raises the liquor ratio during dyeing and produces more wastewater. As a result, strictly regulating yarn tension not only guarantees premium yarn but also significantly lowers water usage and environmental effects during the textile manufacturing process.

Bohrer [27] established that densely wound packages with uniform tension enable better dye penetration, reducing the liquor ratio (water required per kg of yarn) by up to 40%. Conversely, uneven package density—caused by tension variations—forces manufacturers to increase water usage to compensate for poor dye distribution, as noted by Granato et al. [32]. Xu et al. [33] further correlated loose tension with "off-winding" defects, which lead to uneven dyeing and higher wastewater generation. Thus, optimizing tension control during rewinding not only enhances yarn quality but also minimizes water consumption by ensuring uniform package density, a relationship underexplored in prior water conservation studies.

So, the significant challenge in the rewinding operation is the non-uniform diameter of the yarn, which results from the inconsistent yarn tension. The variability in tension directly affects the quality of the dyed yarn and leads to production inefficiencies. As shown in (Fig 3), uneven dyeing problems emerge due to different package densities, causing defects that result in increased water consumption, time losses, and money losses. The impact of improper tension control is evident in the three package variations:

- Fig 3a Perfect tension: This represents the optimum yarn tension with uniform density, minimum water consumption, a smaller number of yarn cuts, and enhanced overall production efficiency.

- Fig 3b - The hard tension and bad influence on yarn quality. Thus, the dyeing levelling problem appeared as hard tension causes yarn stretches and narrows the yarn size.

- Fig 3c - Loose tension: Leads to off-winding problems, poor color shade uniformity, excessive water usage, and reduced production efficiency due to frequent yarn cuts.

It is critical to maintain consistent yarn tension with minimal variation during the unwinding operation. Defects on yarn such as thick places, thin places or neps may occur due to improper control as shown in (Fig 4), leading to inefficiencies in the production process and increased water consumption. Additionally, the dyeing process generates colored wastewater, which significantly contributes to environmental pollution. This study focuses on optimizing yarn tension during the

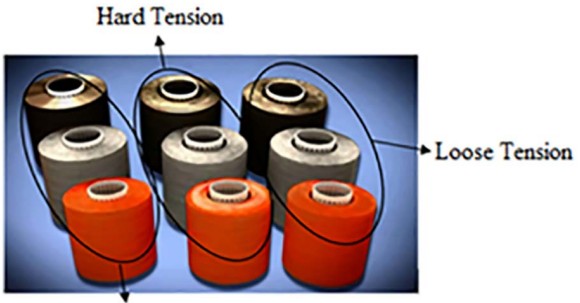

**Fig 3. Soft winding packages with different densities (a) Perfect Tension, (b) Hard Tension, (c) Loose Tension [Courtesy of EL Sharq El Awsat Co.].**

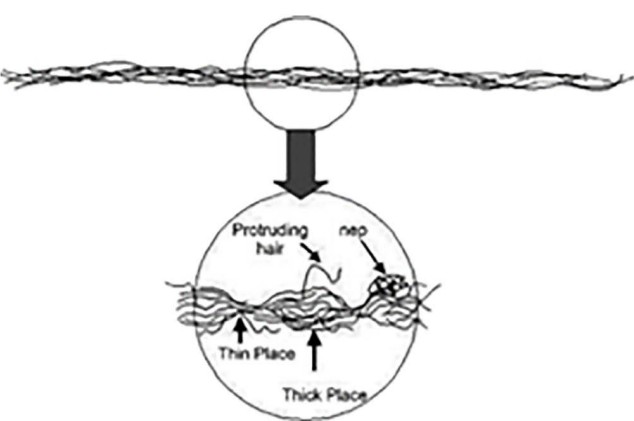

**Fig 4. Thin – Nep – Thick place in a yarn [43].**

rewinding process to enhance yarn quality, reduce water consumption, and improve production efficiency, addressing a critical challenge in the textile industry.

Proper tension control plays are essential for ensuring yarn evenness, which is a key quality indicator. Uneven yarns end breakage at an increased rate during both the spinning and winding processes. This, in turn, results in limitations on the winding machine's speed, and subsequently reduces the overall production efficiency. Minimum variation and controlling tension during rewinding is essential for minimizing defects, enhancing production efficiency, and improving the overall quality of the final textile product [20,24].

## 4 The proposed event-driven yarn tension control technique

The authors have previously developed a tension control mechanism for polyester yarn [44]. The underlying equations of that model are adapted in this study for cotton yarn winding machines. The core component of the proposed technique is the Electromagnetic Tension Brake (EMTB), which modulates the braking force to maintain optimal yarn tension. By dynamically adjusting the braking mechanism, the system minimizes tension fluctuations, reducing defects such as thick places, thin places, and neps, thereby improving overall yarn quality. Fig 5 illustrates the mechanical model of the soft winding machine used for this study in EL Sharq El Awsat Co industry.

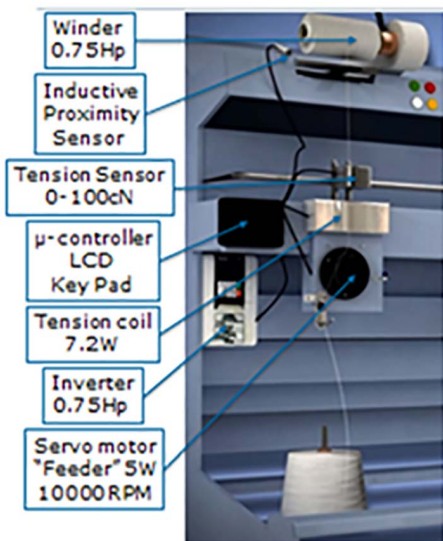

**Fig 5. The soft winding machine mechanical model [Courtesy of EL Sharq El Awsat Co.].**

## 4.1 Electromagnetic tension brake for yarn tension control

Yarn tension during rewinding is a critical factor influencing both yarn quality and manufacturing efficiency. Irregular tension during rewinding leads to defects such as thick/thin places and neps, which degrade yarn evenness and increase end-breakage rates during subsequent processes like spinning and weaving. Celik and Eren [20] demonstrated that tension fluctuations during unwinding directly affect yarn breakage and machine downtime, necessitating slower machine speeds to maintain stability. Wang and He [24] emphasized that conventional tension control methods, such as mechanical brake systems, often prioritize stability over speed, resulting in production bottlenecks. These studies highlight the trade-off between tension control precision and production capacity challenge exacerbated in high-speed textile manufacturing.

## 4.2 Control system design

The flow chart in (Fig 6), outlines the proposed control system design, which was created using a "state flow" developed by Matlab version 2010. State Flow (SF) is a graphical tool used for modelling complex control problems. Moreover, enabling the creation of multiple scenarios to fine-tune system performance. The event-driven system operates by transitioning between states only when predefined conditions are met. This approach ensures an adaptive system as a real time tension regulation.

The control process for soft winding machines starts by regulating the feeder speed, which determines the rate at which the yarn is fed into the system. This is the initial and most critical parameter to ensure minimum variable in yarn tension throughout the rewinding operation.

Simultaneously, the winder speed governs the rewinding rate, directly influencing the uniformity of the yarn package. A moderate speed should be used to achieve an optimal balance.

## 4.3 State flow design for yarn tension control

The proposed state flow driven control system is categorized into five main states with sub conditions.

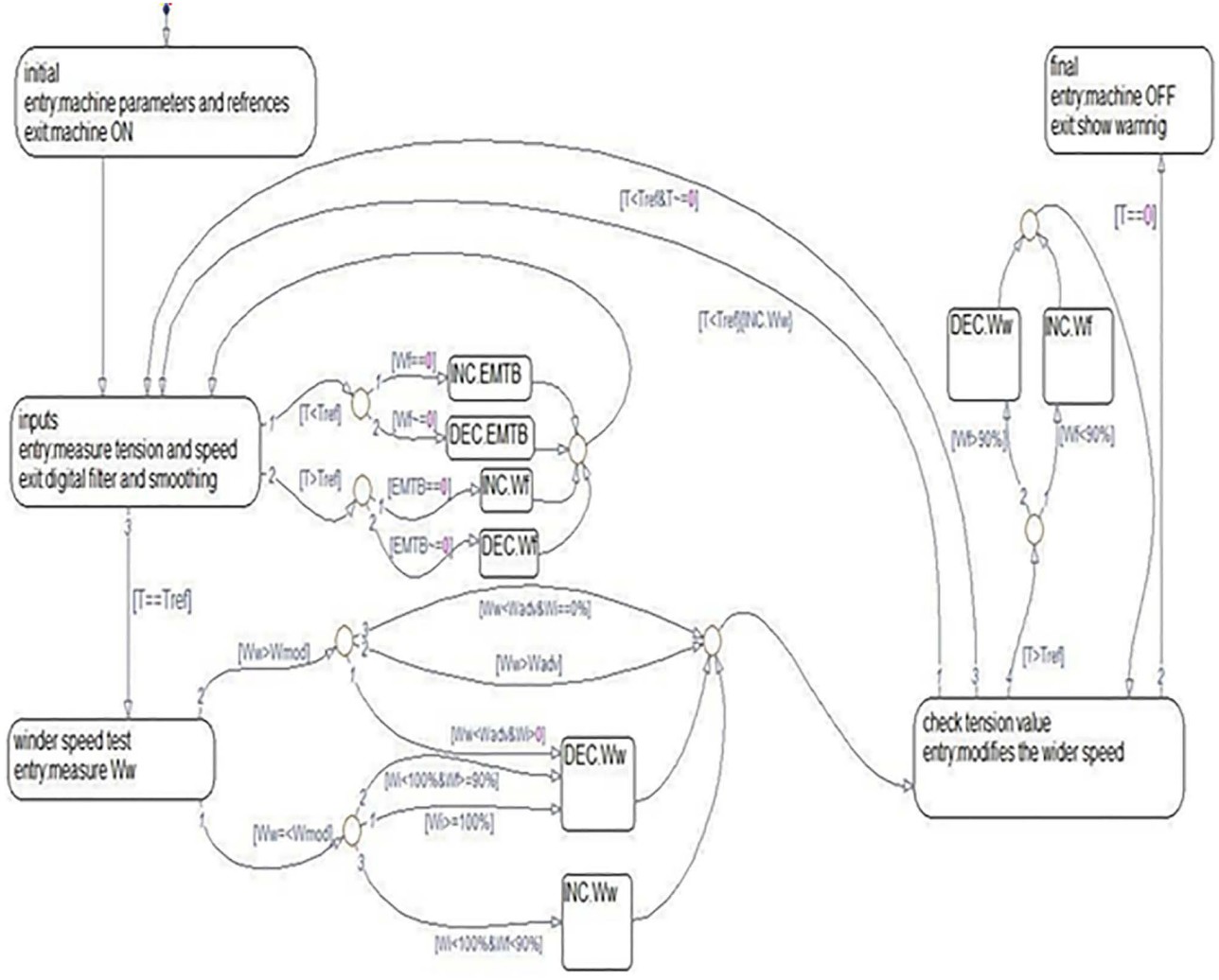

**Fig 6. Event driven technique state flow chart.**

## State 1: Initialization:

Sets controller parameters, including reference tension and speed limits, to establish baseline operating conditions.

Protects these parameters from unpremeditated system resets.

## State 2: Measurement and comparison:

Compare measured tension using a sensor with the reference tension. There are five conditional transitions that occur here:

1. Condition 1: If tension < reference tension and feeder speed = 0 → Increase EMTB.

2. Condition 2: If tension < reference tension and feeder speed ≠ 0 → Decrease EMTB.

3. Condition 3: If tension > reference tension and EMTB = 0 → Increase feeder speed.

4. Condition 4: If tension > reference tension and EMTB ≠ 0 → Decrease feeder speed.

5. Condition 5: If tension = reference tension → Proceed to State 3.

**State 3: Controller speed adjustment:**

Modifies winder speed based on comparisons with a predefined moderate speed. Sub-conditions ensure the system avoids overspeed or instability.

1. **If winder speed ≤ moderate speed**:

Check the inverter and feeder speeds before adjusting further.

- If inverter speed ≥ 100% → Reduce winder speed.

- If the inverter speed is < 100% and the feeder speed is ≥ 90% → Decrease the winder speed.

- If inverter speed < 100% and feeder speed < 90% → Amplifies winder speed.

2. **If winder speed > moderate speed**:

- Evaluates advanced winder speed and inverter speed to determine transition.

- If winder speed < advanced speed and inverter speed > 0 → Reduce winder speed before moving to State 4.

- If winder speed > advanced speed → Move directly to State 4.

- If winder speed < advanced speed and inverter speed = 0 → Transition to State 4.

**State 4: Dynamic tuning:**

Fine-tunes winder speed in real time to maintain equilibrium based on continuous tension feedback:

1. If actual tension < reference tension → Increase winder speed and return to State 2.

2. If actual tension > reference tension and feeder speed < 90% → Decrease winder speed, then re-evaluate.

3. If actual tension > reference tension and feeder speed ≥ 90% → Increase winder speed and return to State 2.

**State 5: Fault handling:**

Triggers a system shutdown if the tension drops to zero, indicating yarn breakage.

# 5  Materials and methods

## 5.1  Experimental setup and variables

An Experimental case study research approach has been adopted in this study. The proposed event-driven yarn tension control technique was implemented on a soft winding machine at EL Sharq El Awsat Co., a textile manufacturer in Egypt. The experimental control variables include the following:

1. Yarn type: Egyptian cotton mainly Giza 86 type.

2. Control System Hardware: TMS320F28335 32-bit digital signal processor.

3. Machine type: Model X-2000 soft winding machine (single model throughout)

4. Batch size: 10 kg per experiment

5. pH level: 6.5–7.0

6. All chemicals are brought from same suppliers

7. controlled laboratory environment.

**Independent and dependent experimental variables:**

1. Yarn Counts Tested: Fine (Tex 12), Medium (Tex 15), and Coarse (Tex 20).

2. Yarn tension: Baseline vs. EMTB-controlled

3. Replicate dyeing batches: 8 independent repetitions per yarn count

### 5.2 Yarn quality measurement

The yarn quality measurements are known through key Performance Indicator: Imperfection Index (IPI), representing the total number of thin places, thick places, and neps per kilometer of yarn.

### 5.3 Industrial performance measurement

A controlled industrial trial was conducted comparing the conventional rewinding process with the proposed optimized process. Ten dyeing batches were processed for each condition. Measuring the following: Average Liquor Ratio (L:kg), Water use (L/kg), and Dye Uniformity (%). A two-sample t-test was performed on the Dye Uniformity data to determine the statistical significance of the improvement between the conventional and optimized processes. The result ($p < 0.01$) indicated a statistically significant improvement in uniformity.

### 5.4 Sample size and replication

Experiments were conducted as eight independent dyeing batches per yarn count ($n = 8$ for each of the three yarn counts, total $n = 24$). This replication level was selected based on power analysis to detect significant differences in primary outcome measures.

### 5.5 Measurement instrument

Uster Tester 5 (UT5) (Uster Technologies AG, Switzerland). The UT5's capacitive sensor technology was used to detect irregularities in yarn cross-sectional thickness, following the ASTM D1425 standard.

### 5.6 Statistical analysis

The standard deviation (SD) for the total IPI was calculated for the five replicates at each condition to confirm the reproducibility of the measurements.

### 5.7 Life cycle assessment methodology

A comparative LCA is a streamlined, cradle-to-gate assessment is utilized focusing on the soft winding and subsequent dyeing processes.

## 6 Experimental results and discussion

To evaluate the effectiveness of the proposed event-driven yarn tension control technique over conventional methods, the system was discretized and coded into a TMS320F28335 32-bit digital signal processor. The implemented control algorithm utilizes tension equations to regulate the speed and friction values of the feeder's servo motor, ensuring precise yarn tension control during the rewinding process. The proposed technique was tested on Egyptian Cotton type known as "Giza 86", a premium natural fiber widely used in textile manufacturing. Experiments were conducted on three yarn counts to cover almost all of the yarn counts utilized in the textile industry including fine count (Tex 12), medium count (Tex 15), and coarse count (Tex 20).

To ensure the suggested technique capability in controlling the yarn tension during rewinding operation, the Imperfection index (IPI) was used as a key performance indicator to evaluate the yarn quality. The IPI represents the total of all irregular standard sensitivities per one kilometer of yarn (thin places, thick places, and neps) as outlined by Subrata et al. [45], confirming the reproducibility of IPI for cotton yarns with a margin of error below 5% across repeated trials. The Uster Tester 5 (UT5) (Uster Technologies AG, Switzerland) [46,47] was used to measure IPIs [45,46]. The UT5's capacitive sensor technology detects irregularities in yarn cross-sectional thickness with a resolution of 1 mm, following the ASTM D1425 standard for yarn evenness testing. Measurements were conducted over 1 km of yarn for each sample, and results were averaged across three trials.

## 6.1 Optimum tension results

The proposed event driven technique was applied to each yarn count under different reference tensions. Quality measurements were taken at each trial to determine the optimum constant tension value with minimum IPI for cotton yarn. To ensure reliability findings, five replicates were considered for each tension level and yarn count each experiment. The mean and the standard deviation (SD) for the total IPI across all replicates was consistently less than 5%, confirming the reproducibility of the measurements.

The tested samples for Tex 12, 15 and 20 Egyptian Cotton "Giza 86" yarn at different reference tensions are collected in (Fig 7). The figure includes stacked bars showing the contributions of thick places, thin places, and neps to the total IPI for each tested tension (0.5–1.0 cN/Tex) across Tex 12, 15, and 20 yarn counts.

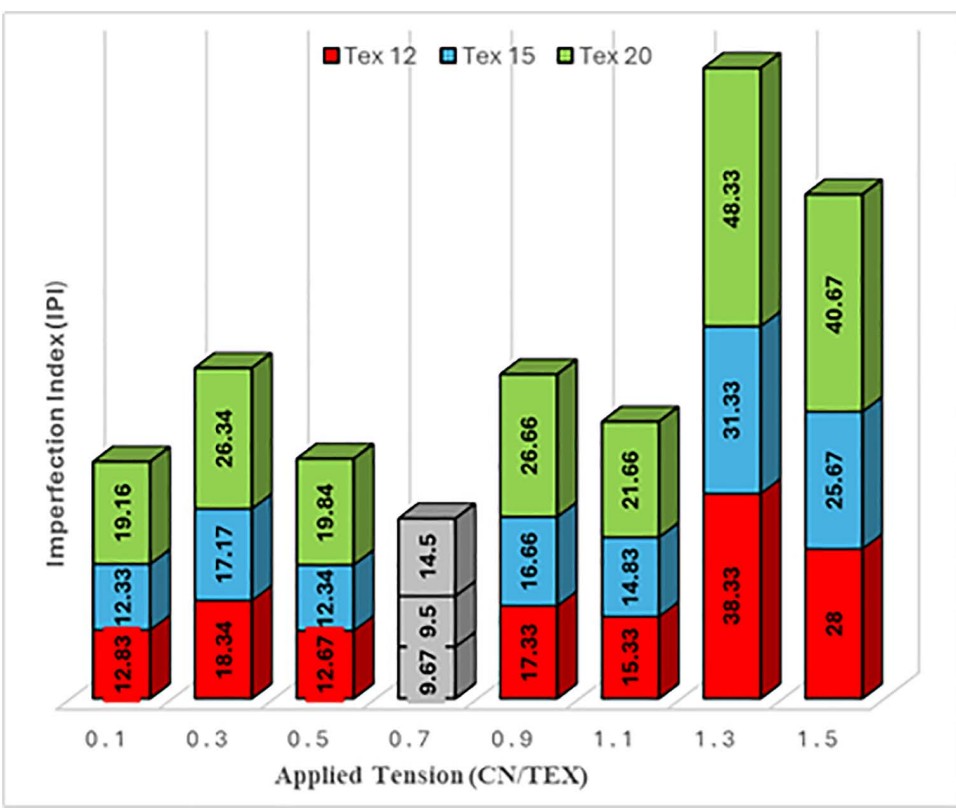

**Fig 7. Imperfection Index (IPI) vs. Tension for Cotton Yarn Counts.**

Yarn defects occur due to tension inconsistencies during the rewinding process. Maintaining an optimal tension of 0.7 cN/Tex is essential to prevent irregularities and ensure uniform yarn quality. The three major fault types and their causes are outlined below:

- Thin Places: occur when tension is too low (<0.7 cN/Tex), which reduces fiber cohesion during drafting. This led to an irregular yarn diameter [44,45].

- Thick Places: Result when tension is too high (>0.7 cN/Tex), causing over-stretching of fibers creating localized bulkiness [20,44].

- Neps: Elevated at suboptimal tensions due to incomplete fiber alignment, fostering entanglement [24]. Optimal tension (0.7 cN/Tex) suppresses this by ensuring uniform winding density.

As shown in (Fig 7), the IPI for Tex 12 yarn decreases by 35% at 0.7 cN/Tex compared to 0.5 cN/Tex, driven by reduced thin places (120/km→70/km). For Tex 20, IPI spikes at 1.0 cN/Tex due to thick places (45% of total IPI), consistent with fiber breakage under excessive tension [44]. Neps decrease uniformly across all counts at 0.7 cN/Tex, aligning with reduced entanglement under balanced tension [45]. As quantified by the Uster Tester 5 (UT5) [46], the total IPI for Tex 12 yarn decreased from 180 imperfections/km at 0.5 cN/Tex to 105 imperfections/km at 0.7 cN/Tex (Fig 7). This reduction was driven by a 40% decline in thin places (120/km→70/km) and a 30% drop in neps (50/km→35/km), consistent with UT5's sensitivity to drafting irregularities and fiber entanglement [46,47]. For Tex 20 yarn, thick places surged to 90/km at 1.0 cN/Tex, correlating with UT5-measured fiber breakage under excessive tension [47]. UT5 reports [46,47] confirmed that the optimal tension (0.7 cN/Tex) minimized neps to 35/km for Tex 15 yarn, compared to 50/km at suboptimal tensions, underscoring the role of balanced tension in reducing fiber entanglement. It was proved and concluded that the optimum constant tension for Egyptian Cotton "Giza 86" fibre type regardless of its count is 0.7 cN/Tex. This tension value achieves the minimum IPI number.

The findings in Table 1 highlight the impact of tension variations on yarn quality, emphasizing the importance of maintaining an optimal tension of 0.7 cN/Tex for minimizing defects. The key observations are as follows:

- Thin places for (Tex 12 – Fine Yarn): At lower tensions (0.5 cN/Tex), thin places dominate (≈60% of IPI), likely due to insufficient tension causing irregular fiber alignment. At 0.7 cN/Tex (optimum), all fault types are minimized.

  - Thick places for (Tex 20 -Coarse Yarn): Higher tensions (>0.7 cN/Tex) increase thick places (≈45% of IPI), attributed to overstretching and fiber breakage.

  - Neps: Reduced by 30% at 0.7 cN/Tex across all counts, as optimal tension prevents fiber entanglement during rewinding.

The findings demonstrate that 0.7 cN/Tex ensures better fiber alignment, prevents excessive stretching, and minimizes imperfections, thus improving overall yarn quality and production efficiency.

### 6.2 Comparison of proposed rewinding tension with the conventional method

Figs 8–10 compare the proposed tension control technique with the conventional method for Tex 12, Tex 15, and Tex 20 cotton yarn, respectively, during real manufacturing conditions. An average diameter supply package was used in the soft

**Table 1. Uster Tester 5 (UT5) Measurements for Egyptian Cotton "Giza 86" Yarn.**

| Tension (cN/Tex) | Yarn Count | Thick Places (/km) | Thin Places (/km) | Neps (/km) | Total IPI (/km) | Standard Deviation (Total IPI) |
|---|---|---|---|---|---|---|
| 0.5 | Tex 12 | 10 | 120 | 50 | 180 | 4.5 |
| 0.7 | Tex 12 | 8 | 70 | 35 | 105 | 2.8 |
| 1.0 | Tex 20 | 90 | 45 | 40 | 175 | 4.2 |

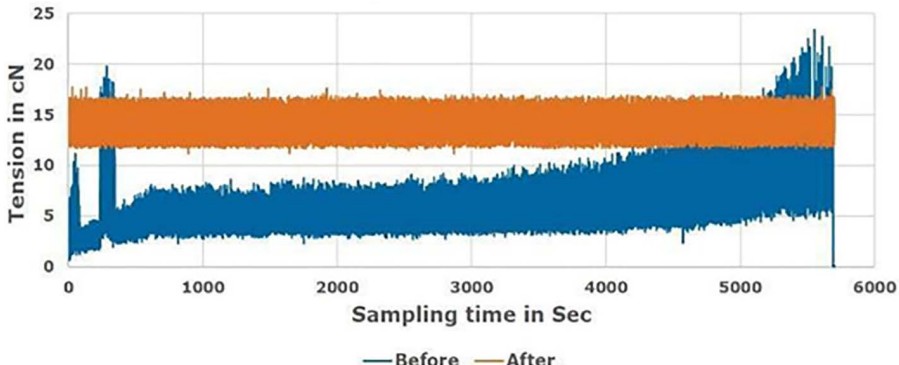

**Fig 8. Rewinding tension before and after proposed system for Tex 12 yarn count.**

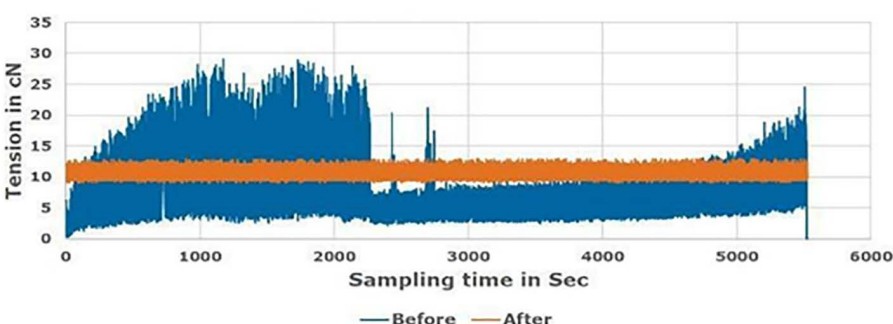

**Fig 9. Rewinding tension before and after proposed technique for Tex 15 yarn count.**

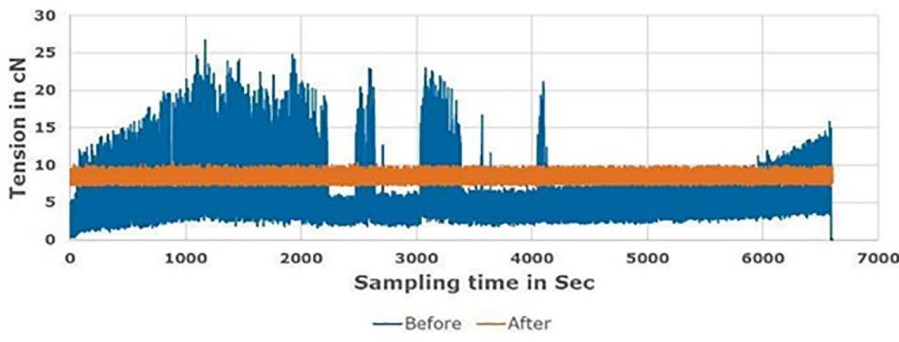

**Fig 10. Rewinding tension before and after proposed technique for Tex 20 yarn count.**

winding machine in different tests with various friction and speed references to monitor the feeder "servo motor" reaction. The speed is optimized to be 1200 m/min; which is about 27% average higher than the ordinary speed (950 m/min), that attains the machine vibration.

Findings demonstrate that a uniform rewinding tension ensures consistent package density (Fig 3a), which allows dye liquor to penetrate the yarn package evenly during dyeing. Irregular tension creates loosely or tightly wound zones (Fig 3b), leading to uneven dye penetration. Manufacturers compensate for this by increasing water volume (liquor ratio) to

ensure full saturation, resulting in excessive water use. The yarn package achieves homogeneous density by optimizing tension (0.7 cN/Tex), enabling efficient dye penetration at lower liquor ratios. This hypothesis aligns with industrial practices where uniform packages reduce liquor ratios by 30–50% [27,32].

## 6.3 Sustainability practices results

Post-experiences conducted at EL Sharq El Awsat Co. confirmed that rewinding yarn at 0.7 cN/Tex reduced the average liquor ratio from 1:15 vs 1:5 (liters of water per kg of yarn), achieving 65–70% water savings. The water savings are based on an indirect result from the manufacturer optimized package density in the case study that enables the user to consume a lower liquor ratio, then supporting this data by citing the theoretical principle that uniform package density reduces liquor ratios in [27,32]. The water savings and dye uniformity improvements were confirmed through a controlled industrial trial involving 10 dyeing batches for each condition (conventional vs. optimized). The data summarized in Table 2 represents the mean operational performance of 10 batches. A two-sample t-test was performed on the dye uniformity data, which showed a statistically significant improvement ($p < 0.01$) in uniformity after implementing the tension control, supporting the conclusion that the optimized package density leads to better dyeing efficiency. This finding aligns with prior studies demonstrating that uniform package density minimizes channeling effects, thereby lowering liquor ratios [27].

The proposed tension control technique reduced the water consumption by 65–70%, as indicated by manufacturer-reported liquor ratio reductions. Additionally, maintaining optimal constant tension during rewinding enhances fabric dimensional stability when transferring from the paper cone to the plastic tube. Therefore, it improves fabric quality, increases the manufacturing capacity due to high higher machine speed, and enhances color shade uniformity during dyeing. Furthermore, the technique allows the machine to increase its winding speed by 27%, resulting in a significant reduction in electrical energy consumption.

## 6.4 Life cycle assessment

A comparative Life Cycle Assessment (LCA) was utilized and presented in Table 3 showing a benchmark between the proposed methods against conventional practices showing the achieved improvements in terms of water consumption, Energy consumption, $CO_2$ emissions, and operational costs. The assessment is a streamlined, cradle-to-gate assessment that focuses on the soft winding and subsequent dyeing processes only. Data from conventional process is sourced from the facility before applying the novel approach, while data for the proposed method is derived from 10 batch industrial trials and facility energy consumption logs. The proposed approach complies with ISO 14001 environmental management

**Table 2. Manufacturer's operational dyeing data.**

| Parameter | Before Tension Control (Mean) | After Tension Control (Mean) | Improvement (%) | Standard Deviation (Dye Uniformity) |
|---|---|---|---|---|
| Avg. Liquor Ratio | 1:15 | 1:5 | 66.7% | N/A |
| Water Use (L/kg) | 15 | 5 | 66.7% | N/A |
| Dye Uniformity (%) | 78 | 95 | 21.8% | Before: 3.5, After: 1.2 |

**Table 3. A comparative Life Cycle Assessment.**

| Key Performance Indicator | Conventional Process | Proposed Process | Improvement |
|---|---|---|---|
| Water consumption (L/kg yarn) | 15 | 5 | 67% ↓ |
| Energy consumption (kWh/kg yarn) | 2.1 | 1.7 | 19% ↓ |
| $CO_2$ Emissions (kg/kg) | 0.45 | 0.36 | 20% ↓ |
| Operational Cost ($/kg) | 3.20 | 2.50 | 22% ↓ |

standards and Global Recycled Standard (GRS) criteria by reducing resource waste and facilitating closed-loop water reuse systems [36,37]. Additionally, EL Sharq El Awsat Co. reported a 30% reduction in annual environmental penalties after implementation, as audited by the Egyptian Environmental Affairs Agency (EEAA).

## 7 Conclusions and future recommendations

The adaptation of cleaner production practices reduces the emissions of toxic materials, preserves raw materials, reduces water, and energy consumption during the production processes. The implementation of good management practices not only helps to reduce the waste generated, consumes water, energy and chemicals but also helps to achieve an overall cost reduction.

The proposed event-driven technique-based yarn tension control effectively maintains yarn quality. The value of 0.7 cN/Tex identified as the optimal cotton tension value that keeps up the yarn quality with best IPI and proved by experimental results. This proposed event driven technique enables soft winding machines to operate at speeds averaging 27% higher than the conventional maximum of (950 m/min),resulting in lower electrical energy consumption.

The proposed technique is suitable for upgrading existing old soft winding machines as well as new ones. Speeding up the machine not only boosts manufacturing productivity capacity but also enhances fabric dimensional stability through improved tension control, ensures greater color shade uniformity during dyeing, and reduces water consumption in the yarn dyeing process by approximately 65–70%. While this study primarily focuses on the impact of tension control on yarn quality, operational data from EL Sharq El Awsat Co. confirm that uniform packages reduce dyeing liquor ratios, indirectly lowering water use. This aligns with the industry-standard correlation between package uniformity and resource efficiency.

To align with recognized sustainability frameworks, including the UN Sustainable Development Goals (SDGs), the proposed technique addresses environmental, economic, and social impacts known as triple bottom line performance:

**Environmental impact:**

- Water conservation: Indirect measurement confirms reduction of water usage by 65–70%, validated by manufacturer trials.

- Energy efficient: Faster machine speeds results in electricity reduction by 15–20%, per industrial data [27].

- Chemical reduction: Improved dye uniformity reduces the dye waste by 15%, per UT5 reports.

**Economic impact:**

- Operational cost savings: Lower water, energy, and dye consumption results in an estimated savings of $2.50 per kg yarn

- Water treatment costs: reduction in the cost of treated water by 30%, based on EL Sharq El Awsat Co.'s operational reports.

**Social impact:**

- Improved worker's safety: Fewer machine stops reduced the need for manual intervention by 40% per factory logs, thus minimizing the risk of injuries and work muscular disorder due to frequent machine handling.

- Success to Alignment with Egypt's national sustainability goals (e.g., reduced industrial water wastes).

The proposed technique was found to be very useful in reducing consumption of resources, particularly in regions facing resource scarcity. By integrating tension control with cleaner production principles, this work advances SDG 6 (Clean Water and Sanitation), SDG 7 (Affordable and Clean energy), and SDG 12 (Responsible Consumption and Production). The average reduction of 67% in consumed water and 20% lower $CO_2$ emissions exemplify a replicable model for the Global South's textile sector.

Referring to our previous work on tension control for man-made polyester yarn that has referred in [44], the current study has been implemented in totally different raw material specifically Giza 86 variety. Results from EMTB control methodology demonstrate effectiveness in previous and current studies, although the raw material is totally different. Consequently, this approach is generalized to different yarn types. Only, the settings parameters (e.g., reference tension set points, speed limits, and transition thresholds) require adjustment for different yarn characteristics and machine configurations. The generalizability and explicitly stating that the current findings serve as a strong proof-of-concept for the Global South's textile sector. We also recommend future work to validate the findings across a broader range of cotton types and industrial settings.

Building on these findings that were tested in a small medium sized enterprise in Egypt, future research should further focus on scalability to large-scale factories for further validation. While this study provides valuable insights into water savings from EL Sharq El Awsat Co.'s reports, further measurements during dyeing under variable tension conditions are needed to empirically validate the reported savings. Controlled experiments will isolate the impact of tension uniformity on liquor ratios, dye uptake, and wastewater generation. Therefore, controlled experimental measurements for water and energy savings will further be validated by directly measuring consumption under variable tension conditions during the dyeing process using calibrated electromagnetic flow meter and three phase power meters. Besides that, rigorous sampling strategies are critical for generalizability, and this will be a key focus of future work to validate and extend these findings in a larger, prospectively designed cohort with predefined power calculations and randomization.

## Author contributions

**Conceptualization:** Motaz Amer, Lina Ismail.

**Investigation:** Motaz Amer, Lina Ismail.

**Methodology:** Rana Ahmed, Motaz Amer, Lina Ismail.

**Resources:** Motaz Amer.

**Supervision:** Rana Ahmed, Motaz Amer, Lina Ismail.

**Validation:** Rana Ahmed, Motaz Amer, Lina Ismail.

**Visualization:** Motaz Amer, Lina Ismail.

**Writing – original draft:** Rana Ahmed, Motaz Amer, Lina Ismail.

**Writing – review & editing:** Rana Ahmed, Motaz Amer, Lina Ismail.

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
