## [Decision Letter · Decision Letter 0]

17 Nov 2025

PONE-D-25-48765A Sustainable Yarn Tension Control Technique for Optimizing Textile Dyeing Efficiency and Water UsePLOS ONE

Dear Dr. Amer,

Thank you for submitting your manuscript to PLOS ONE. After careful consideration, we feel that it has merit but does not fully meet PLOS ONE’s publication criteria as it currently stands. Therefore, we invite you to submit a revised version of the manuscript that addresses the points raised during the review process.

We look forward to receiving your revised manuscript.

Kind regards,

Maria H Ribeiro, PhD

Academic Editor

PLOS ONE

Journal Requirements:

4. Please ensure that you refer to Figures 4 and 11 in your text as, if accepted, production will need this reference to link the reader to the figures.

Reviewers' comments:

Reviewer's Responses to Questions

**Comments to the Author**

1. Is the manuscript technically sound, and do the data support the conclusions?

Reviewer #1: Partly

Reviewer #2: Yes

2. Has the statistical analysis been performed appropriately and rigorously? 

Reviewer #1: No

Reviewer #2: N/A

3. Have the authors made all data underlying the findings in their manuscript fully available?

Reviewer #1: Yes

Reviewer #2: Yes

4. Is the manuscript presented in an intelligible fashion and written in standard English?

Reviewer #1: No

Reviewer #2: No

5. Review Comments to the Author

Reviewer #1: Overall

According to the journal guidelines (https://journals.plos.org/plosone/s/submission-guidelines#loc-manuscript-organization), the manuscript organization should be:

Title page: List title, authors, and affiliations as first page of the manuscript

Abstract

Introduction

Middle section

The following elements can be renamed as needed and presented in any order:

Materials and Methods

Results

Discussion

Conclusions (optional)

Ending section

Acknowledgments

References

Supporting information captions (if applicable)

The manuscript is not organized according to this set of guidelines, this must be adressed

The authors should summarize the state of the art and highlight the relevance of the work submitted in the Introduction section. In the present form it is too broad, and not fully focused, e.g., the authors digress too much on general matters related to cleaner production, the introduction should be more focused

Some statistical treatment of the data is strongly advised, to highlight if the observed differences are statistically meaningful, e.g., for data in Table 2, how many replicates were performed, what was the standard deviation of the water use, and variation in dyeing uniformity?

The authors refer and present data (Table 3) of a life cycle analysis (LCA), but they do not present detailed information in materials and methods, e.g., scope, system boundaries, inventory data used, software used, uncertainty and sensitivity, which ISO guidelines were followed

Minor glitches: “Furthermore, water Demand” should be “Furthermore, water demand”

Mind that reference 14 is the pdf version of a powerpoint presentation prepared by the Institute of Environmental Engineering, Kaunas University of Technology, Lithuania and sponsored by the Sponsored by UNEP, Division of Technology, Industry, and Economics, with the title: INTRODUCTION TO CLEANER PRODUCTION (CP) CONCEPTS AND PRACTICE. The authors should consider using as reference a peer-reviewed paper focused also on cleaner production goals, strategies and implementation, e.g., DOI: 10.1002/csr.2693, https://doi.org/10.3390/su15032161, …

Reviewer #2: 1. The authors’ claim of achieving 65–70% water savings and approximately 20% energy reduction during dyeing appears to rely primarily on manufacturer-reported liquor ratios rather than controlled experimental measurements, which undermines the validity of the conclusion.

2. The sections discussing environmental impact and wastewater characteristics are well referenced; however, they are not clearly connected to yarn tension control, resulting in a diluted and unfocused contribution.

3. The materials and methods section requires clearer and more systematic description to ensure reproducibility and transparency.

4. The authors are strongly encouraged to provide directly measured water and energy consumption data, rather than inferring savings solely from liquor ratio values.

5. It must be clarified whether dyeing recipes, dye classes, batch sizes, and machine types were held constant across all experiments, as these variables can significantly influence the results.

6. The manuscript should report the number of dyeing batches tested, along with variance measures and statistical significance to support the reliability of the findings.

7. The IPI analysis appears to be based on only three trials per yarn count, yet no confidence intervals or statistical tests are provided to evaluate consistency or uncertainty.

8. Although the authors suggest broad industrial applicability, the study was conducted solely on Egyptian cotton (Giza 86) within a single manufacturing facility, limiting the generalizability of the conclusions.

9. The manuscript frequently blends prior literature, factory-reported data, and experimentally measured results without clear distinction, which affects objectivity and makes it difficult to assess the source and reliability of specific claims.

6. PLOS authors have the option to publish the peer review history of their article (what does this mean?). If published, this will include your full peer review and any attached files.

Reviewer #1: No

Reviewer #2: No

---

## [Author Response · Author response to Decision Letter 1]

29 Jan 2026

The authors would like to seize this chance to thank the editor and reviewers for their appreciated effort and time. The authors respond to all the raised comments, attempting to fulfill all the requirements through this stage. The authors hope that their reply receives considerable satisfaction and also would like to clarify their sincere intention to make every effort/modification required to attain the reviewers’ satisfaction and raise the quality of the submitted manuscript to fit with the journal's high standards.

---

## [Decision Letter · Decision Letter 1]

23 Apr 2026

A Sustainable Yarn Tension Control Technique for Optimizing Textile Dyeing Efficiency and Water Use

PONE-D-25-48765R1

Dear Dr. Motaz Amer,

We’re pleased to inform you that your manuscript has been judged scientifically suitable for publication and will be formally accepted for publication once it meets all outstanding technical requirements.

Kind regards,

Maria H Ribeiro, PhD

Academic Editor

PLOS One

Additional Editor Comments (optional):

Reviewers' comments:

Reviewer's Responses to Questions

**Comments to the Author**

1. If the authors have adequately addressed your comments raised in a previous round of review and you feel that this manuscript is now acceptable for publication, you may indicate that here to bypass the “Comments to the Author” section, enter your conflict of interest statement in the “Confidential to Editor” section, and submit your "Accept" recommendation.

Reviewer #2: All comments have been addressed

6. Review Comments to the Author

Reviewer #2: (No Response)

7. PLOS authors have the option to publish the peer review history of their article (what does this mean?). If published, this will include your full peer review and any attached files.

Reviewer #2: No

---

## [Editor Report · Acceptance letter]

PONE-D-25-48765R1

PLOS One

Dear Dr. Amer,

I'm pleased to inform you that your manuscript has been deemed suitable for publication in PLOS One. Congratulations! Your manuscript is now being handed over to our production team.

Kind regards,

on behalf of

Dr. Maria H Ribeiro

Academic Editor

PLOS One